# Realizing string-net condensation: Fibonacci anyon braiding for universal gates and sampling chromatic polynomials

Zlatko K. Minev [1,8], Khadijeh Najafi[1,2,9], Swarnadeep Majumder[1,9], Juven Wang [3,4], Ady Stern [5], Eun-Ah Kim [6,7]✉, Chao-Ming Jian [6]✉ & Guanyu Zhu[1]

The remarkable complexity of a topologically ordered many-body quantum system is encoded in the characteristics of its anyons. Quintessential predictions emanating from this complexity employ the Fibonacci string net condensate (Fib SNC) and its anyons: sampling Fib-SNC would estimate chromatic polynomials while exchanging its anyons would implement universal quantum computation. However, physical realizations remained elusive. We introduce a scalable dynamical string net preparation (DSNP) that constructs Fib SNC and its anyons on reconfigurable graphs suitable for near-term superconducting processors. Coupling the DSNP approach with composite error-mitigation on deep circuits, we create, measure, and braids Fibonacci anyons; charge measurements show 94% accuracy, and exchanging the anyons yields the expected golden ratio $\phi$ with 98% average accuracy. We then sample the Fib SNC to estimate chromatic polynomial at $\phi + 2$ for several graphs. Our results establish the proof of principle for using Fib-SNC and its anyons for fault-tolerant universal quantum computation and aim at a classically hard problem.

In principle, complex quantum many-body states can efficiently encode solutions to provably hard computational problems. However, protocols to generate such states, coveted even for intermediate-scale systems, without relying on random gates, remain elusive[1]. Surprisingly, a state of topological quantum matter considered for fault-tolerant universal quantum computation could also harbor such solutions. Specifically, a fascinating connection between a string-net condensate wave function supporting Fibonacci anyons[2,3] and a classically hard problem of evaluating chromatic polynomials has been noted[4–7].

String-net condensates (see Fig. 1a) are many-body vacuum states, encompassing essentially all parity- and time-reversal-invariant topological phases[2]. A string-net condensate is a complex superposition of

'string nets', which can be visualized as trivalent graphs representing spins in an excited state $|1\rangle$ or ground state $|0\rangle$ and subject to local geometric rules. The simplest condensate whose geometric rules allow a string to 'branch' into two strings is the Fibonacci string-net condensate (Fib-SNC) (Fig. 1b)[2,3]. This simple 'branching rule' for the Fib-SNC nevertheless leads to a remarkably complex state when combined with '$F$-move' rules (see Fig. 1c), which mandate intricate relationships among the superposition amplitudes, dictated with the golden ratio $\phi$. In this state, the modulus-squared amplitude of a string-net is determined by[4–7] the chromatic polynomial[8] of its dual graph evaluated at $\phi + 2$. Evaluating the chromatic polynomial is generally #P-hard and even its approximation is computationally hard[9–13]. Note that for counting problems, #P is the analog of the more familiar class NP for

[1]IBM Quantum, T.J. Watson Research Center, Yorktown Heights, Newyork, NY, USA. [2]MIT-IBM Watson AI Lab, Cambridge, MA, USA. [3]Center of Mathematical Sciences and Applications, Harvard University, Cambridge, MA, USA. [4]London Institute for Mathematical Sciences, Royal Institution, London, UK. [5]Department of Condensed Matter Physics, Weizmann Institute of Science, Rehovot, Israel. [6]Department of Physics, Cornell University, Ithaca, NY, USA. [7]Department of Physics, Ewha Womans University, Seoul, South Korea. [8]Present address: Google Quantum AI, Mountain View, California, USA. [9]These authors contributed equally: Khadijeh Najafi, Swarnadeep Majumder. ✉e-mail: eun-ah.kim@cornell.edu; chao-ming.jian@cornell.edu

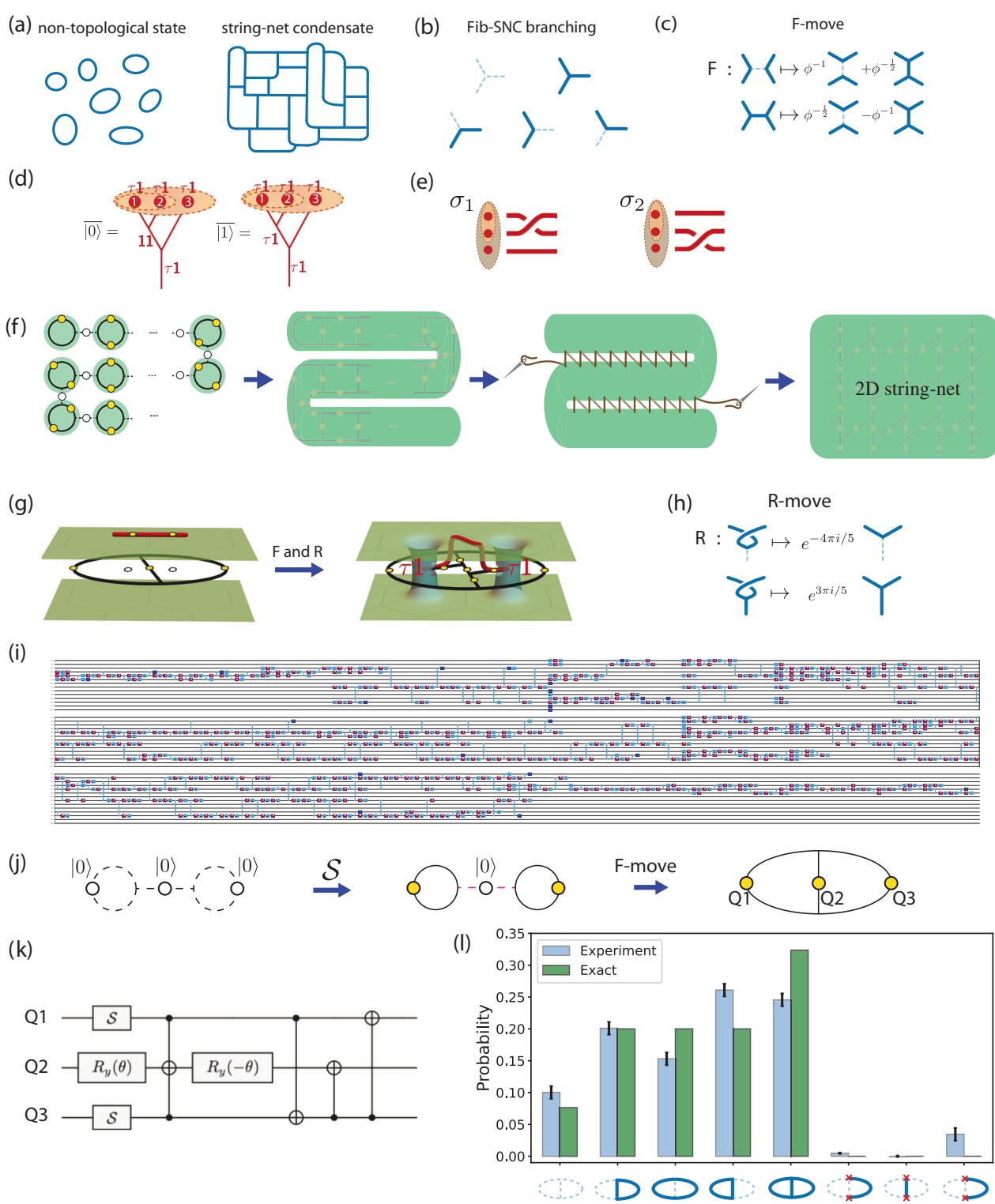

decision problems. Hence, realizing the Fib-SNC could open a door to a new class of classically-hard problems.

Another complexity of Fib-SNC is its role as a vacuum state supporting emergent anyons, capable of fault-tolerant universal quantum computation[2,3]. The underlying universal nature of these anyons is captured by a long-wavelength effective theory treatment[2], which combines two time-reversed copies of a topological quantum field theory (TQFT)[14–16]. In principle, even a single copy allows for universal quantum computation[17], but no microscopic blueprint exists for manifesting just a single copy. Each copy resembles the TQFT

proposed for filling-factor 12/5 quantum Hall state[18] and supports an anyon type $\tau$, whose multi-anyon Hilbert space dimension follows the Fibonacci sequence. This arises from a 'fusion rule' (see Fig. 1d) where two $\tau$ anyons brought together can either annihilate (resulting in **1**) or fuse into a single $\tau$. The 'doubling' leads to three anyons types, $\tau$**1**, **1**$\tau$, and $\tau\tau$[19], with Fib-SNC providing a microscopic blueprint. A triplet of any of these anyons can encode a logical qubit, capitalizing on the two possible fusion outcomes of the $\tau$. For instance, realizing Fib-SNC and creating two pairs of $\tau$**1** allows for encoding logical information to the fusion outcome of the first two $\tau$**1** anyons (see Fig. 1d)[20]. Exchanging

**Fig. 1 | Principle of the dynamical string-net preparation (DSNP) approach and experiment. a** Typical string-net configurations of spins in $|1\rangle$ state, for a trivial state (left) and for a string-net condensate (right). **b** Branching rule for the Fib-SNC. A dashed line represents a qubit in the $|0\rangle$ state while a solid line represents a qubit in the $|1\rangle$. **c** Five-qubit F-move relating allowed string-net configurations among five qubits. When one or two pairs of four outer legs are identified, this becomes four-qubit or three-qubit F-moves. **d** Logical qubit encoding using a triplet of $\tau\mathbf{1}$ anyons. Logical $\overline{|0\rangle}$ and $\overline{|1\rangle}$ differ by the fusion outcomes of the first two $\tau\mathbf{1}$ anyons. The red lines in the figure represent the space-time trajectory of anyons. **e** Pairwise braiding among the triplet of $\tau\mathbf{1}$ anyons implements a non-Clifford gate on the logical state encoded on the triplet of anyons(see SM Sec. I A). **f** Schematic outline of DSNP. Yellow dots represent qubits in SNC, and empty dots represent reserve qubits in $|0\rangle$

state. (see SM Sec. IV for details). **g** The Fib-SNC can be visualized with a pair of two-dimensional surfaces representing two time-reversed copies of TQFT. To create two $\tau\mathbf{1}$ anyons, we bring in an open string from above. F- and R-moves bring the ends of the open string to join the two copies of TQFT through wormholes, with the ends piercing the wormholes and localizing anyons. **h** The R-moves (or its complex conjugate) to resolve the over-crossing (see SM Sec. I A for more details). **i** A deep quantum circuit for braiding two $\tau\mathbf{1}$ anyons using hardware-native gates (see Fig. 3). **j** DSNP for the smallest Fib-SNC. **k** Quantum circuit corresponding to two $S$ gates followed by the 3-qubit F-move, implementing all the steps of panel (**j**). **l** The probability weight of different string-net configurations corresponding to the depicted string nets for the minimal Fib-SNC. The red × marks the vertices violating the branching rule. Only 5% of the shots violate the branching rule.

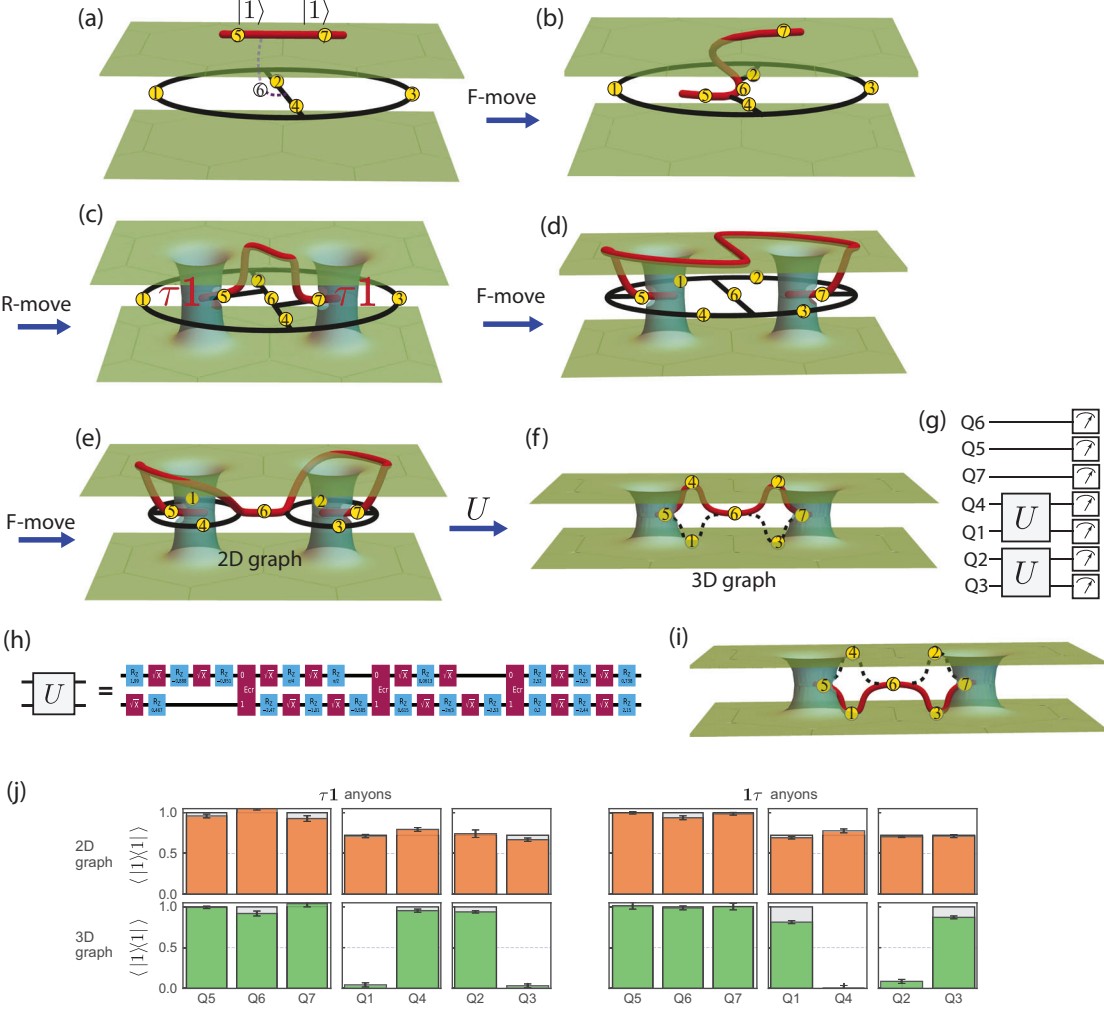

**Fig. 2 | Fibonacci anyons: creating anyon pairs and certifying their anyon charges. a–c** Building a minimal two-plaquette string-net and creating a pair of $\tau\mathbf{1}$ anyons. **a** Qubits Q1--Q4 are initialized in $|0\rangle$ encoding an empty two-plaquette graph, Q5 and Q7 initialized in $|1\rangle$ support the open string in the upper sheet. Q6 in $|0\rangle$ is an ancillary unoccupied string segment. **b** An F-move transforms the graph into a 3D configuration. **c** An R-move on Q6 yields a pair of $\tau\mathbf{1}$ anyons localized on the left and right plaquettes, with Q5 and Q7 acting as their tail qubits. The resulting open string threads through the upper plane, with its two ends supporting the $\tau\mathbf{1}$ anyons that pierce the two wormholes in the sheets (recall Fig. 1g). One further applies F-moves flipping edge Q2 and Q4 to reach the configuration in (**d**). **d, e** The anyon type is diagnosed by a charge measurement performed using a further F-move flipping edge Q6 to deform the graph into two connected plaquettes (**e**),

joined at Q6. **f** Unitaries $U$ act on pairs (Q4, Q1) and (Q2, Q3) to place Q4 and Q2 on the open string. **g** The circuit for the anyon type certification. **h** The compilation of the unitary $U$ with three 2-qubit ECR gates (See SM Sec. VI) and a few single-qubit gates for the 2-qubit unitary $U$. **i** The contrast experiment prepares a pair of $\mathbf{1}\tau$ anyons by replacing the R-move with its complex conjugate. Thus, in the final measurement stage, the open string goes through the bottom plane with qubits Q1 and Q3 in its path. **j** Theory (gray bars) vs. experiment (colored bars) for the $\tau\mathbf{1}$ (left) and $\mathbf{1}\tau$ (right) anyon pairs measured in the 2D graph in panel (**e**) before applying $U$(upper) and the 3D graph in panel (**f**) after applying $U$(lower). The expectation values $\langle |1\rangle\langle 1| \rangle$ are indistinguishable in the 2D graph. However, in the 3D graph, the $|1\rangle$-state expectation values of unpinned qubits Q1--4 certify the anyon type as the measurement reveals the path of the open string.

the anyons, braiding their space-time trajectories, enacts non-Clifford logical gates, a primary requisite for universal quantum computation (see Fig. 1e).

Unfortunately, faithfully realizing the Fib-SNC and its anyons has been unreachable, despite successful implementation of topological states with Abelian anyons[21,22] and even non-Abelian Ising[23,24] and $D_4$ anyons[25], whose braiding is restricted to Clifford gates at best. Conventionally, the Fib-SNC is viewed as the ground state of a static Hamiltonian on a hexagonal lattice marked by high-order, 12-spin interactions[2], which are exceptionally difficult for current capabilities. Nevertheless, a recent experiment showed promise[26]. Yet, the formidable circuit depths necessary for $F$-moves forced the use of approximations. Moreover, the need to control 12 qubits for the smallest plaquette makes exploring the condensation of graph configurations practically infeasible.

## Results
### The DSNP strategy
Our approach to creating minimalistic Fib-SNC is the scalable dynamic string-net preparation (DSNP) strategy (see Fig. 1f). As explained below, we implement this strategy to create $\tau\mathbf{1}$ anyons, confirm their anyon charges, and braid them to extract the golden ratio. Furthermore, DSNP allows us to make the first steps towards scaling up the Fib-SNC to estimate the chromatic polynomial at the golden ratio $\phi + 2$.

DSNP leverages the inherent flexibility of graphs for efficient dynamical preparation of the Fib-SNC (see Fig. 1f). This is in contrast to the proposal of operating on a rigid lattice[26,27]. A similar graph-centric perspective has proven productive for preparing states with Ising anyons[28]. A single physical qubit can represent the smallest isolated string-net, or 'bead', when prepared in a valid superposition through the modular-$\mathcal{S}$ gate:

$$\mathcal{S} = \frac{1}{\sqrt{1 + \phi^2}} \begin{pmatrix} 1 & \phi \\ \phi & -1 \end{pmatrix}. \tag{1}$$

The next step in building a larger-scale Fib-SNC is to insert a qubit initialized in the $|0\rangle$ state between beads. Using $F$-moves (see Fig. 1c), the beads can be entangled into a strip of plaquettes. This strip can then be folded and sewn into a two-dimensional Fib-SNC through additional $F$-moves. The dynamic nature of this process allows for the optimization of resources for specific aims. The depth of the circuit grows linearly with the system size, the best scaling expected for a unitary circuit preparation of a topologically ordered state[29], but with the smallest prefactor to our knowledge compared to previous proposals, such as ref. 27. To prepare the Fib-SNC with $N \times N$ plaquettes, ref. 27 estimated $\approx 120N$ layers of parallel CNOTS. With DSNP, the required depth scales as $2N$ 5-qubit $F$-moves. Using a conservative estimation of 40 CNOTs per 5-qubit $F$-moves[27], the depth of the DSNP scales as $80N$ CNOT layers, with all other steps being parallelizable.

Creation of anyons must change the topology of the many-body state. In order to establish a protocol that allows explicit association between the circuit implementation and the evolving topology of the many-body state, we introduce a three-dimensional (3D) graphical representation where each copy of the TQFT is depicted as a two-dimensional surface (see Fig. 1g). While the anyon-free Fib-SNC can be visualized entirely through two-dimensional (2D) graphs, the creation of anyons whose 'anyon charge' labels which of the two copies of TQFT is affected necessitates keeping track of the two copies. Anyons connect the two surfaces through 'worm holes' at the locations of anyons. Furthermore, to create anyons while allowing for the detection and correction of local errors, we follow the 'tail anyon' strategy[20] that traps the end of an open string to the 'tail qubit' located on a dangling edge

inside a plaquette. The $\tau\mathbf{1}$ or $\mathbf{1}\tau$ pair-creation can now be visualized as bringing in an open string from above or below the two surfaces. Fig. 1g illustrates inserting the strings from above into the Fib-SNC state shared between the two surfaces, which requires undoing the overcrossing using the '$R$-move' shown in Fig. 1h.

As a flexible state preparation strategy built on graphs, DSNP allows the preparation of Fib-SNC with an arbitrary number of plaquettes. The smallest of such only requires three qubits forming two plaquettes, which can be prepared as shown in Fig. 1j using a circuit with two $\mathcal{S}$ gates and a three-qubit $F$-move shown in Fig. 1k (see "**Methods**" for more details). Fig. 1l shows the experimental result of implementing this Fib-SNC on the 27-qubit IBM Falcon processor *ibm_peekskill*. Using dynamical decoupling and readout-error mitigation[30], but without other error mitigation, we sample the probability distribution of computational bitstrings using 8192 experimental shots. The x-axis labels represent bitstrings as their corresponding graph configurations, with dashed (solid) lines indicating qubits in the $|0\rangle$ ($|1\rangle$) state and red × 's denoting broken strings. Full tomography reconstruction of the experimental state yields a fidelity of $0.87 \pm 0.01$ to the ideal state, which is not high in the absolute sense. However, the state shows a much higher degree of 95% adherence to the branching rule.

### Anyon creation and certification
Now we create a pair of Fibonacci anyons and certify their anyon types in the above two-plaquette Fib-SNC state. To create the $\tau\mathbf{1}$ pair, we introduce an open string (red) of qubits (Q5 and Q7) above the two-plaquette Fib-SNC (Fig. 2a). With an unoccupied string (Q6 initialized in state $|0\rangle$) as the bridge, we entangle the open string with the Fib-SNC via $F$-moves and an $R$-move as illustrated in Fig. 2a–c (see "**Method**" for details). These moves restore the planarity of the graph and effectively create two wormholes connected by an open string through the upper-copy TQFT. Now, both plaquettes each host a $\tau\mathbf{1}$ anyon at the tail. Although the qubits, except the tail qubits, now respect the local rules of Fib-SNC, the two copies of the TQFT share a complex superposition through the wormholes. To create the other anyon type, the $\mathbf{1}\tau$ anyons, the open string should be inserted from underneath the two-plaquette Fib-SNC rather than from above. Practically, this amounts to using the conjugate $R^*$-move instead of the $R$-move.

The canonical way to certify the anyon type would be to measure the five-qubit plaquette operators for each plaquette in the two-dimensional graph in Fig. 2c. However, to combat the noise obscuring the certification in such extended measurements, we introduce an alternative approach that reduces this certification to independent single-qubit measurements. We first deform the graph so that the open string is pinned in the middle to be shared between the two TQFT's, as shown in Fig. 2e. Now, each plaquette can be independently measured, while the two tail qubits (Q5 and Q7) and the qubit bridging the plaquettes (Q6) are fixed to be in the $|1\rangle$ state, irrespective of the anyon type $\tau\mathbf{1}$ or $\mathbf{1}\tau$. At this point, all the qubits are still shared between the two TQFT's and we referred to this state as "2D graph". We then lift the remaining four qubits off the shared space through a basis-changing unitary represented as $U$. In the end, the open string passes through all but two of the qubits. In particular, measurements of lifted qubits (Q1, Q4, Q2, Q3) in the final "3D graph" shown in Fig. 2f for $\tau\mathbf{1}$'s amount to measuring the open string itself with a definite parity associated with the anyon types (see "**Method**" for details on the implementation of Fig. 2e, f).

To experimentally realize the anyon pair preparation and the anyon charge measurements, we need high-accuracy circuits about 150 two-qubit-gate-layers deep. We use a 133-qubit IBM Heron processor *ibm_torino*, featuring fast gates and reduced cross-talk, with median single- and two-qubit gate fidelities of $3.6 \times 10^{-4}$ and $4.6 \times 10^{-3}$, respectively (see SM Sec. VI). To address experimental noise, we employ a composite error suppression and mitigation strategy,

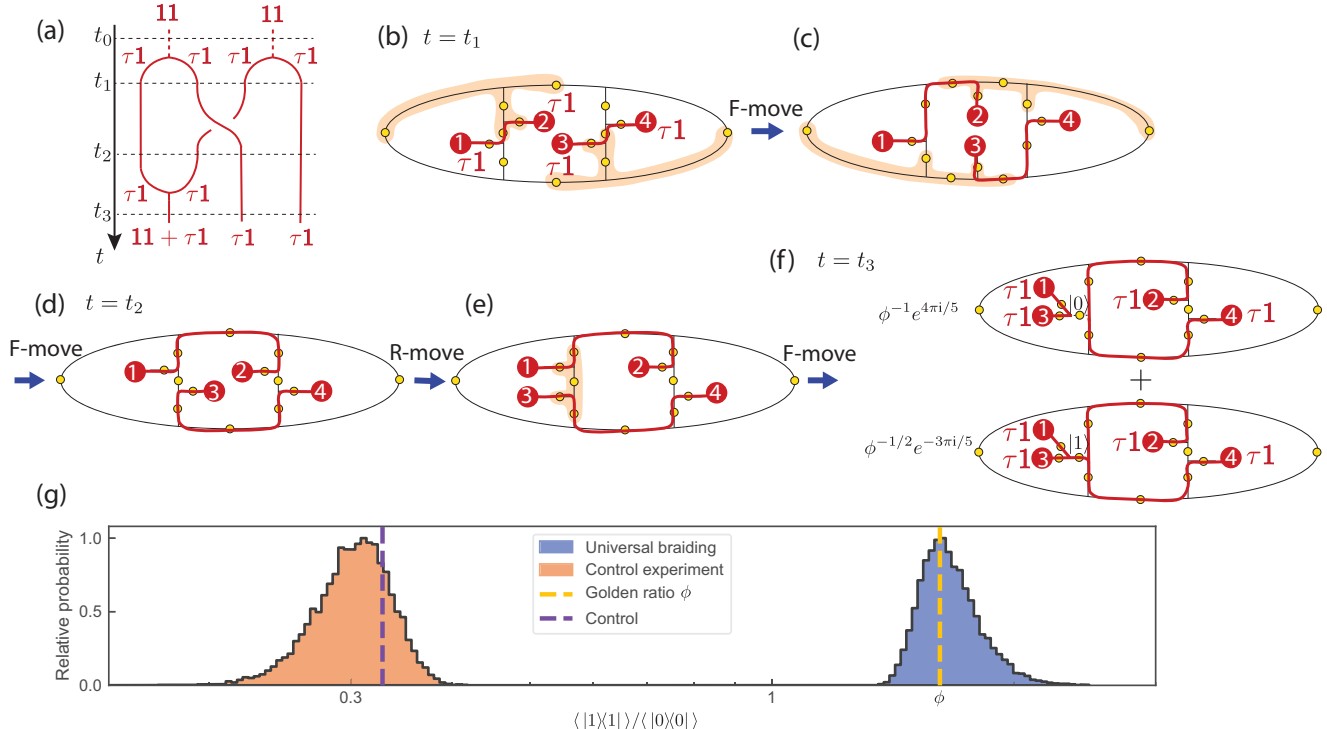

**Fig. 3 | Braiding of two $\tau\mathbf{1}$ anyons to implement a non-Clifford gate on a topological logical state. a** Worldlines depicting the creation of four $\tau\mathbf{1}$ anyons from the vacuum $\mathbf{11}$, followed by the braiding of anyons 2 and 3, and concluding with a fusion-based measurement to determine the logical gate implemented by the braiding process. **b** Generalization of the protocol from Fig. 2, where four $\tau\mathbf{1}$ anyons (red dots labeled as 1–4) are initialized on three plaquettes. Labels for the qubits (yellow dots) are suppressed. **c**, **d** Braiding is achieved through four five-qubit F-moves, which permute anyons 2 and 3. The groups of five qubits undergoing the F-moves are indicated by the orange patches. **e** An R-move flips anyon 3 from the center to the left plaquette, forming a new configuration for fusion. **f** A final F-move fuses anyons 1 and 3, resulting in a coherent superposition of two fusion outcomes. **g** Experimental distributions of the measured ratio $\langle 1|1\rangle/\langle 0|0\rangle$ on a logarithmic scale, derived via bootstrap resampling for both the braiding and control experiments. The analysis accounts for error mitigation and statistical uncertainty (see SM Sec. VIII). Vertical dashed lines indicate theoretical predictions: the golden ratio $\phi$ (yellow) and the control value (purple). Asymmetry in the distributions reflects the non-linear transformation of the ratio observable.

including real-time qubit selection, dynamical decoupling, twirling[31,32], zero-noise extrapolation[33–35], and twirled readout-error mitigation[30] (see SM Sec. VII).

We now create pairs of each anyon type and certify their type through single qubit measurements over $8.8 \times 10^6$ experimental realizations across 1100 quantum circuit instances (see SM Sec. VIII). In Fig. 2j shows the resulting measurement outcome statistics for each of the qubits corresponding to the two anyon types. We measured 2D graphs like Fig. 2e and 3D graphs like Fig. 2f. The measurement results shown in Fig. 2j confirm the prediction with high precision. Specifically, three pinned qubits Q5–7 are consistently measured in the $|1\rangle$ state at all times. In the 2D graph, although single-qubit measurements hide the anyon type even for the remaining four qubits (Q1–4), the measured expectation value of $\langle 1|1\rangle$ of $0.73 \pm 0.04$ as shown in the upper histograms in Fig. 2j is consistent with the theoretically predicted value of $\frac{\phi^2}{\phi^2+1} \approx 0.72$. However, these four qubits (Q1–4) show a dramatic contrast between the two anyon types in 3D graphs, as shown in the lower histograms in Fig. 2j. Since these four qubits are "lifted off" the shared plane to belong to top or bottom TQFT, the open string traverses the top TQFT with Q4 and Q2 ($\tau\mathbf{1}$) or the bottom TQFT with Q1 and Q3 ($\mathbf{1}\tau$) depending on the anyon type. (see **Method** for more details).

## Braiding doubled Fibonacci anyons

Fully two-dimensional braiding must involve three or more plaquettes and two pairs of $\tau\mathbf{1}$ anyons. DSNP prescribes a scalable strategy for creating plaquette strips of arbitrary lengths. In Fig. 3, we

demonstrate two-dimensional braiding in a scalable and error-correctable manner using the minimalistic three-plaquette strip and verify the braiding outcome through the fusion of a pair of anyons (see Fig. 3a for the schematics). Repeating the anyon pair preparation, we prepare two anyon pairs spread over three plaquettes as depicted in Fig. 3b. This amounts to time steps $t_0$–$t_1$ in Fig. 3a. Initially, the logical qubit encoded to the triplet of $\tau\mathbf{1}$ anyons (1,2,3) is in the $\overline{|0\rangle}$ state since the anyon 1 and anyon 2 are created from vacuum. Now, we braid $\tau\mathbf{1}$ anyons 2 and 3 using a sequence of exact F-moves executing the time steps $t_1$–$t_2$ in Fig. 3a. Such braiding is predicted to execute a non-Clifford gate $\sigma_2$ (see Fig. 1e) on the logical qubit, rotating the logical state to

$$\sigma_2\overline{|0\rangle} = \phi^{-1}e^{4\pi i/5}\overline{|0\rangle} + \phi^{-1/2}e^{-3\pi i/5}\overline{|1\rangle}. \quad (2)$$

We certify the predicted non-Clifford gate by fusing anyon 1 and anyon 3. For this, we bring anyon 1 and anyon 3 together to share a single root edge using an R-move and an F-move(see Fig. 3e). Now a measurement in the physical computational basis of the root edge onto either $|0\rangle$ or $|1\rangle$ projects the logical qubit to $\overline{|0\rangle}$ or $\overline{|1\rangle}$, respectively. Hence, if the braiding implements the correct logical gate in Eq. (2), the golden ratio can be measured through $\langle 1|1\rangle/\langle 0|0\rangle = \phi$.

As in the previous experiment, we implement this sequence on *ibm_torino* using the composite mitigation strategy, but with double the number of twirls and shots per twirl due to the increased circuit complexity. We find $\langle 1|1\rangle/\langle 0|0\rangle = 1.65 \pm 0.14$, within 2% of the golden ratio $\phi$. Figure 3g shows the distribution of bootstrap

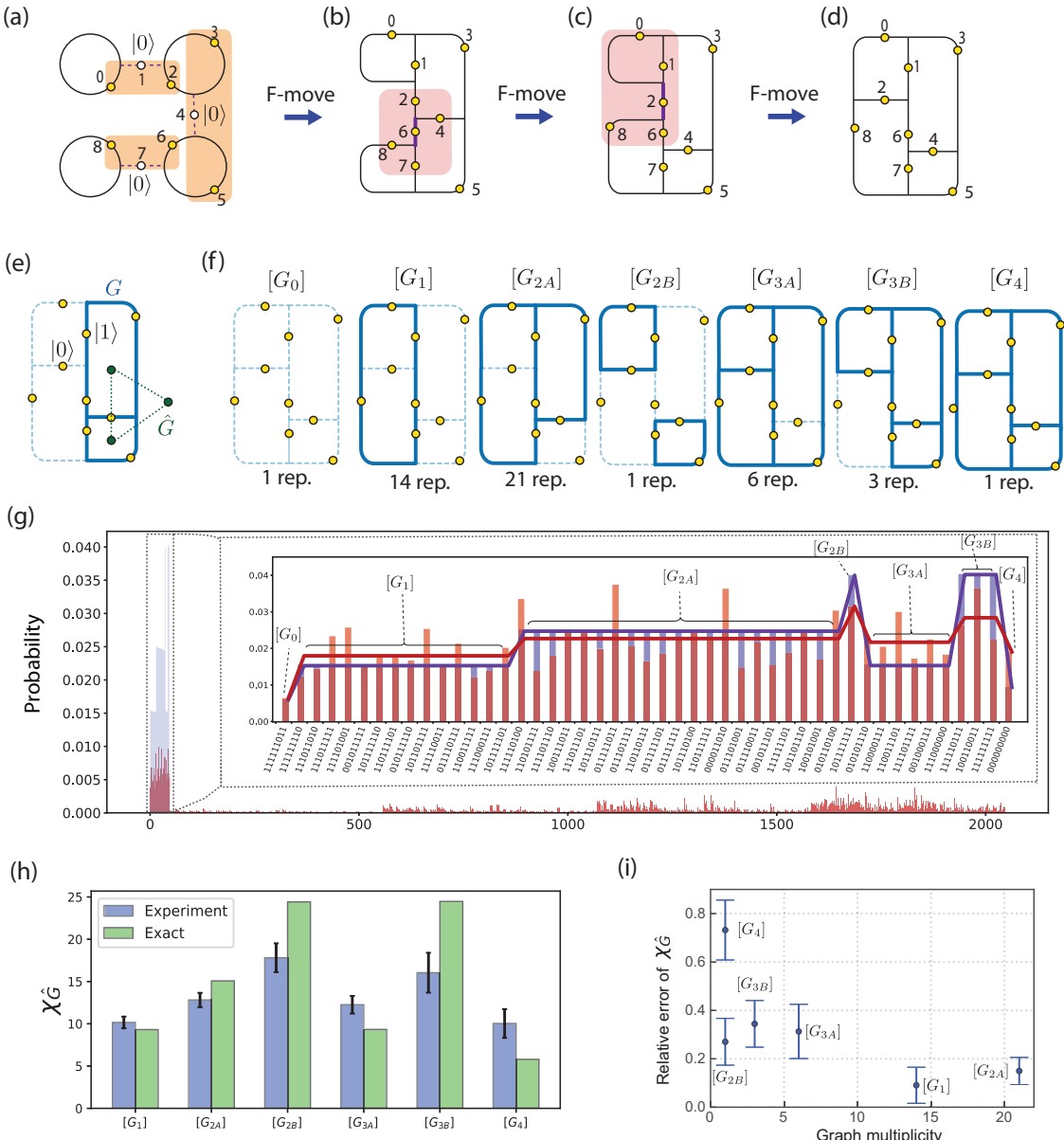

**Fig. 4 | Estimating chromatic polynomials. a, b** Four decoupled beads are prepared by generalizing the protocol of Fig. 1j. Three parallel *F*-moves act on the three shaded groups of qubits (orange boxes), yielding a folded strip with 4 plaquettes. **c, d** Two 5-qubit *F*-moves applied to qubits in the shaded boxes deform the graph into a 2 × 2 lattice of plaquettes: a 2D Fib-SNC. **e** The dual graph of each graph *G* is denoted as $\hat{G}$ (green dotted lines). **f** For the 2 × 2 lattice, there are 7 isomorphism classes of graphs formed by the edges in state |1⟩. All the graphs are topologically equivalent (or isomorphic) within each isomorphism class. The number of representative graphs (multiplicity) is listed for each isomorphism class. The relative probability with respect to empty configuration $G_0$ is defined as $\bar{P}([G]) = P([G])/P([G_0])$. **g** Large panel: Probability distribution over all

$2^{11}$ bit-strings, including the two ancilla qubits (blue: theory; red: experiment). Theoretically, non-zero bitstrings (47 in total) are ordered on the left. These satisfy the branching rule, while the remaining bitstrings on the right do not. Inset: zoom-in to bitstrings obeying the branching rule. The theoretical distribution reflects 7 isomorphism classes. Thick red line: the measured probability averaged over each isomorphism class. **h** Extracting the chromatic polynomial values for graphs dual to the given string-net isomorphism class (blue: experiment; green: theory). Error bars obtained from the standard deviation of the graph representatives in each class. **i** The relative error and multiplicity of each isomorphism class of graphs. A class with a larger multiplicity tends to have smaller relative errors.

resampling, providing confidence intervals (see SM Sec. VIII). In a control experiment, we intentionally introduce bit-flip errors during to break two strings, generating unwanted excitations (see SM Sec. VIII). This modification alters the bitstring distribution. We now measure ⟨1|⟨1|⟩/⟨0|⟨0|⟩ = 0.30 ± 0.025, consistent with the theoretical prediction of 0.328 for the modified circuit.

**Estimating the chromatic polynomials via string-net sampling**

Now we move onto the most ambitious pursuit of this paper, taking the first step towards a new class of classically hard problems. In Fig. 4, we

realize a two-dimensional, four-plaquette Fib-SNC vacuum and sample it to estimate the chromatic polynomials for all possible trivalent graph embedding. We perform all experiments on *ibm_torino*. Due to the outstanding challenge of mitigating noise for sampling rather than expectation values[35], we only mitigate readout errors and not gate errors. We use DSNP as illustrated in Fig. 4a–d, starting with a four-bead strand and evolving each bead into one of the four-plaquettes of the resulting Fib-SNC vacuum (see Fig. 4d). To reduce the circuit depth of *F*-moves we use 2 ancilla qubits (see SM Sec. VIII), in addition to the 9 qubits participating in Fib-SNC.

It has long been predicted that the normalized probability weight of a subgraph $G$ in Fibonacci string-net condensate evaluates the chromatic polynomial of a dual graph $\hat{G}$ (see Fig. 4e) at $k = \phi + 2$ [4–7], i.e.,

$$\frac{P(G)}{P(G_0)} = \frac{1}{\phi+2}\chi(\hat{G}, \phi+2),\qquad(3)$$

where $P(G)$ and $P(G_0)$ are probability weight of a subgraph $G$ and the empty configuration $G_0$, respectively. While the chromatic polynomial $\chi(\hat{G}, k)$ for a positive integer $k$ counts the number of ways to $k$-color the graph $\hat{G}$ [8], a recurrence relation defining the polynomial allows for extension of the polynomial to non-integer valued $k$, such as $\phi + 2$. As a complex combinatorial problem, the evaluation or estimation of the chromatic polynomial is a classically hard problem [9–13] despite the simplicity of the defining recurrence relation. Note that the proof of refs. 11,12 is carried out for rational $k$, while one may expect that the same conclusion holds for irrationals. This implies that the exact theoretical evaluation of the Fib-SNC amplitude requires an exponential-time classical algorithm in general (see SM Sec. III C). Hence the experimental realization of the Fibonacci string-net condensate may offer a new route for seeking quantum advantage.

Although the absence of an error-mitigation scheme poses a challenge in sampling a general state that is not highly concentrated, we can exploit the topological structure of Fib-SNC. Firstly, valid bitstring configurations that satisfy the branching rules form a relatively small subset of all possible bitstrings. Secondly, these valid bitstrings further group into topologically equivalent isomorphism classes. Specifically, for the four plaquette Fib-SNC we implemented, there are 6 classes as shown in Fig. 4f with different multiplicity among the bitstrings that correspond to the class. Figure 4g shows the result of sampling this Fib-SNC vacuum for $30 \times 10^6$ realizations. With two ancilla qubits introduced to reduce the circuit depth, the probability distribution is shown over $2^{11}$ possible bit strings obtained on *ibm-torino*. Leveraging that the Fib-SNC amplitudes can be calculated for the present scale Fib-SNC, we benchmark experimentally sampled results against the exact predictions. The topological nature of Fib-SNC predicts amplitudes of bitstrings to be non-zero only for 47 branching rule respecting bitstrings, with the same amplitude within given isomorphism class (shown in blue in Fig. 4h).

The experimentally obtained probability distribution shows robust suppression of branching-rule violating, forbidden bit-strings (Fig. 4g). Moreover, class averages of the allowed bit-strings offer the estimates of the chromatic polynomials:

$$\chi([\hat{G}_1], k) = k^2 - k \qquad(4)$$

$$\chi([\hat{G}_{2A}], k) = k^3 - 3k^2 + 2k \qquad(5)$$

$$\chi([\hat{G}_{2B}], k) = k^3 - 2k^2 + k \qquad(6)$$

$$\chi([\hat{G}_{3A}], k) = k^4 - 6k^3 + 11k^2 - 6k \qquad(7)$$

$$\chi([\hat{G}_{3B}], k) = k^4 - 5k^3 + 8k^2 - 4k \qquad(8)$$

$$\chi([\hat{G}_4], k) = k^5 - 9k^4 + 29k^3 - 39k^2 + 18k, \qquad(9)$$

at $k = \phi + 2$. For this, we estimate the relative probability $P(G)/P(G_0)$ in Eq. (3) by $\overline{C}([G])/\overline{C}(G_0)$, where $\overline{C}([G])$ represents the average count of all bitstrings corresponding to graphs topologically equivalent to $G$. For a larger scale estimation, a graph class with higher multiplicity can be used as a reference in place of the empty configuration in Eq. (3) (see SM Sec. III E). We show the resulting estimates of the chromatic

polynomial in Fig. 4h, where uncertainty ranges are computed using the standard deviation within each equivalent class. While the absence of error mitigation limits the accuracy of the estimates, Fig. 4i shows the multiplicity within each class countering errors. Specifically, the larger the multiplicity, the more accurate the estimates are. In particular, the experimental estimate based on the average over $[G_1]$-class bitstrings yields 1.82 for the golden ratio $\phi$, with 13% relative error.

## Discussion

In summary, our work introduces and implements a new scalable approach, DSNP, to preparing Fib-SNC and creating, certifying, and braiding the Fibonacci anyons the Fib-SNC supports. While the present experiments with physical qubits are limited by noise, the prospect of sampling of Fib-SNC with a large number of plaquettes using DSNP raises new potential frontier in the pursuit of quantum advantage. The success in this new pursuit of quantum advantage will hinge on solving two open problems. Firstly, noise in sampling must be countered. Secondly, the sampling complexity of Fib-SNC needs to be further investigated to compare it with the complexity of classical approximations. While the sampling space of valid bitstrings still grows exponentially with system size, an estimation of $\chi(\hat{G}, \phi+2)$ for a specific dual graph $\hat{G}$ requires sampling only one equivalence class among the valid bitstrings. Hence, it is possible that the estimation of chromatic polynomial at $\phi + 2$ from quantum sampling can be more efficient than from classical algorithms for intermediate system size (for $O(100)$ to $O(1000)$ qubits). If so, Fib-SNC sampling will become an exciting avenue for near-term quantum computers.

## Methods

### DSNP protocol to create the smallest Fib-SNC

Here, we provide more details on the DSNP protocol for the smallest Fib-SNC as shown in Fig. 1j. We start with three qubits, each prepared in $|0\rangle$ (white dots), representing three unoccupied strings (dashed lines). Then, two single-qubit modular $S$ gates on Q1 and Q3 create two decoupled beads (solid rings). The qubit Q2, which is still in state $|0\rangle$, represents an unoccupied edge between the two beads. Finally, a 3-qubit $F$-move creates the minimal Fib-SNC with two plaquettes by entangling the middle qubit Q2 with Q1 and Q3. The quantum circuit that implements all the steps described above is shown in Fig. 1k.

### Protocol for the anyon creation and certification experiment

In this section, we provide more details about the experimental protocol of creating and certifying doubled Fibonacci anyons presented in Fig. 2. We grow the string-net condensate to create $\tau\mathbf{1}$ and $\mathbf{1}\tau$ anyon pairs and measure their anyon charges. Applying the DSNP concepts from Fig. 1f–g on our minimal example, we add the minimal number of needed qubits, Q4–Q7 shown in Fig. 2a. Qubit Q4 is incorporated into the condensate by entangling it with Q2 using a controlled-NOT operation. Tail qubits Q5 and Q7 are prepared in $|1\rangle$ (yellow dots) and bridge qubit Q6 in $|0\rangle$ (white dot). Now Q1–Q4 form a Fib-SNC shared between two copies of TQFT as depicted in Fig. 2a. To bring in $\tau\mathbf{1}$ anyons, we start by introducing open string Q5–Q7 (red line) above Q2–Q4 (black line). Q6, initially in state $|0\rangle$, is viewed as an ancillary vacuum-string segment (thin dashed line) connecting the plaquettes to the open string. A five-qubit $F$-move (see Fig. 1c) acting on qubits Q2 and Q4–Q7 entangles Q5 and Q7 with the rest, creating a single connected, 3D graph (Fig. 2b). Then, an $R$ move (see Fig. 1h) completes the preparation of a pair of $\tau\mathbf{1}$s at the end of red open string that pierces the wormholes in Fig. 2c. Operationally, using instead a conjugate $R^*$ move related to the $R$ move in Fig. 1h by complex conjugation of phases will amount to creation of $\mathbf{1}\tau$ anyon pairs. In this case, the open string Q5–Q7 should be initially introduced underneath Q2–Q4 instead. At this point, all the qubits except the tail qubits once again follow the local rules of Fib-SNC, forming a complex superposition

shared between the two copies of TQFT. This adherence to the local rules means this state can be error-corrected.

How do we most robustly certify the creation of anyons associated with this abstract notion of an open string? While the qubits are participating in the superposition of string nets as in Fig. 2c, the only measurement predicted to yield a definitive answer would be the left and right five-qubit plaquette operators, each comprising many Pauli terms — but this compounds noise. However, if the qubits are 'lifted' to each copy of TQFT through basis changes such that the open string goes through the qubits, the open string itself can be measured. To this end, we dynamically reconfigure the graph to place Q6 on a bridge that is forced to be in a definite $|1\rangle$ state (see Fig. 2d–e). Now three qubits are pinned in the $|1\rangle$ state: Q6 and the two tail qubits, Q5 and Q7. However, at this point of Fig. 2e, Q1–Q4 will be participating in the condensate, each in a superposition of $|0\rangle$ and $|1\rangle$. At the same time, the open string is away from the qubits except at the wormholes. Remarkably, a change of basis through two-qubit unitary $U$ (see Fig. 2g and SM Sec. II) would lift off Q1–Q4 to make the open string go through Q4 and Q2. Topologically, the two states in Fig. 2e, f are equivalent. Nevertheless, the microscopic qubit placements are such that measurements of Q4 and Q2 will reveal the open string itself with a definite outcome in Fig. 2f.

We certify $\tau\mathbf{1}$ pair creation through the detection of the open string as shown in Fig. 2j (see SM Sec. VIII). The qubits that should be 'pinned' to $|1\rangle$ state, Q5–Q7, are measured to be in the correct state with high probability irrespective of whether the remaining qubits, Q1–Q4, are placed in 2D or 3D graphs, yielding $0.99 \pm 0.05$ on average. However, the measurement outcome for Q1–Q4 are strikingly different because 3D graph has the open string going through (Q1, Q3)-pair or (Q4, Q2)-pair. When placed in the 2D graph with the open string away from the qubits Q1–Q4, the four qubits are indistinguishable, with the expectation value of the one-state projector $\langle|1\rangle\langle1|\rangle$ of $0.73 \pm 0.04$ across the 8 measurements. This is precisely as they should be as a part of the Fib-SNC represented by the 2D graph, with the predicted expectation value of $\frac{\phi^2}{\phi^2+1} \approx 0.72$. On the other hand, when the four qubits are placed in the 3D graph, the open string traverses (Q4, Q2)-pair in the upper copy while the unoccupied string "traverses" (Q1, Q3)-pair in the lower copy. Contrastingly, when $\mathbf{1}\tau$ anyons are prepared with the open string passing through the lower surface, measurement outcomes between (Q4, Q2)-pair and (Q1, Q3)-pair completely reverse, as expected from the open string now passing through (Q1, Q3)-pair in the lower layer. Our certification of $\tau\mathbf{1}$ and $\mathbf{1}\tau$ anyon creation through 28 measurements of the open string show the average experimental discrepancy is $-0.01 \pm 0.06$.

## Data availability
The data supporting the findings of this study are available in the following link: https://zenodo.org/records/15565975.

## Code availability
The codes supporting the findings of this study can be found in the following link: https://zenodo.org/records/15565975.

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

## Acknowledgements

While preparing our manuscript, we learned of a related study by ref. 26 on Fibonacci anyons. One significant difference in our study is that we focus on scalable planar (2D) braiding in an error-correctable manner. Secondly, we introduce the chromatic polynomial estimation through string-net sampling. We thank Sergey Bravyi and Vojtěch Havlíček for insightful discussions on the complexity of chromatic polynomials and Dimitry Maslov for advice on simplifying multi-qubit Toffoli gates. We are grateful to Abhinav Kandala, Emily Pritchett, and Sarah Sheldon for their comments on the manuscript. We also thank Antonio Mezzacapo, Javier R. Moreno, and Ian Hincks for their valuable input. J.W. is supported by the Harvard University CMSA research associate fund. A.S. was supported by grants from the ERC under the European Union's Horizon 2020 research and innovation program (Grant Agreements LEGOTOP No. 788715), the DFG (CRC/Transregio 183, EI 519/71), and the ISF Quantum Science and Technology (2074/19). E.-A.K. acknowledges support by the NSF through OAC-2118310. C.-M.J. is supported by the Alfred P. Sloan Foundation through a Sloan Research Fellowship. G.Z. is supported by the U.S. Department of Energy, Office of Science, National Quantum Information Science Research Centers, Co-design Center for Quantum Advantage (C2QA) under contract number DE-SC0012704.

## Author contributions

Z.K.M., K.N. and S.M. conducted the experiments on the IBM devices, developed the error mitigation protocols and analyzed the data. J.W., A.S., E.-A.K., C.-M.J. and G.Z. developed the theory and protocols and analyzed the data. All authors contributed to the writing of the article.

## Competing interests

The authors declare no competing interests.
