## [Transparent Peer Review file · Nature Communications]

Realizing string-net condensation: Fibonacci anyon braiding for universal gates and sampling chromatic polynomials

Corresponding Author: Professor Eun-Ah Kim

Version 0:

Reviewer comments:

Reviewer #1

(Remarks to the Author)
See the attached report.

Reviewer #2

(Remarks to the Author)
Please see the attached file.

Reviewer #3

(Remarks to the Author)
This is excellent work and deserves publication in a high-profile journal. The research is technically advanced, and the authors have made a commendable effort to make it accessible to a broader audience. I have little to add and recommend the publication as is.

I believe mentioning Ref. 26 in the acknowledgments is sufficient. Since the work appears to have been completed around the same time, multiple references in the main text could potentially confuse the reader.

While I appreciated the opening of the abstract, I should point out that even the vacuum of a harmonic oscillator is not truly empty, as it exhibits quantum fluctuations. The key distinction, in my view, lies in the complexity and many-body nature of topologically ordered vacua.

Reviewer #4

(Remarks to the Author)
The authors use IBM quantum processors to analyse topological behaviour in and around the Fibonacci string-net ground state. In particular, they confirm the intriguing property that the ground-state wavefunction amplitudes depend on the chromatic polynomial. They also show to good accuracy that the anyonic excitations exhibit the non-Abelian braiding properties expected.

Whereas I tend to be skeptical of papers utilising quantum computers to confirm previously understood physics, I think here the physics is rather novel and far from obvious. Moreover, the work done in this paper really pushes the limit of what is experimentally possible -- I would have guessed we would need to wait some years before such intricate properties of Fibonacci anyons could be analysed in a quantum computer. I thus recommend publication of this paper in Nature Communications.

The authors do need to make some small revisions.

-- The first sentence of the abstract is a bit gratuitous, since I don't believe any good physicists are unaware that the ground

state of a quantum system can have non-trivial properties. (Even though particle theorists use the word "vacuum" to mean "ground state", they know it's not trivial.) So I would delete that sentence and not say "topologically ordered vacuum" in the next, but rather "topologically ordered ground state"

-- The authors need to make clear that they are evaluating the chromatic polynomial at a particular number of colours, not in general. They say this in some places, but not everywhere (e.g. in the overview before equation (1) and in their summary).

-- It's not clear what a "tail qubit" is, unless one reads the supplemental information. The authors should add a sentence explaining in the main paper.

-- The pdf of the supplemental information is missing page 6 (at least the one I received).

Version 1:

Reviewer comments:

Reviewer #1

(Remarks to the Author)

We went through the revised manuscript and the authors' responses to the referees' reports. The authors have spent great efforts to improve the manuscript substantially. All our comments/suggestions have been satisfactorily addressed. We are happy to recommend its acceptance in Nature Communications.

Reviewer #4

(Remarks to the Author)

I think the revisions improve meaningfully an already-good paper. I recommend publication in Nature Communications.

Responses to the Referees

We are encouraged that all four reviewers enthusiastically support the publication of our work in a high-profile journal. All four referees enthusiastically endorse the paper in its significance and timeliness. Referees' recommendations were confined to improving clarity and writing.

We have taken this opportunity to improve the polish of the paper following the reviewer's suggestions. Revisions focus on improving clarity and presentation and are limited to the main text. The changes are marked in purple. Overall, the summary of changes is as follows:

- The title now follows the recommendation by Referee 2.
- The first sentence of the abstract has been removed following the recommendation by Referees 3 and 4. We revised what used to be the second sentence per the referees' suggestions.
- We updated Figs. 2 and 3 to enhance the clarity of the labelings to avoid potential confusions identified by Referee 1.
- We fixed the technical issue that led to the omission of a critical page in the supplementary material.
- We introduced a Methods section and moved technical details for creation and certification of anyons to the Methods section to improve the presentation of the novel contributions.
- We have revised the text significantly to improve the quality of writing and to further emphasize the key contribution of this work. Specifically, paragraphs [9], [10], [12], [15-19] have been revised.
- We have clarified the definition of the "tail qubits".

Below, we have compiled the referees' comments point-by-point in blue, and our responses in black.

Report of Referee 1

In this work, the authors explore the quantum simulation of the Fibonacci string-net condensation (Fib-SNC) and demonstrate its potential applications in topological quantum computing. The research is conducted through a series of experiments on the IBM Heron quantum processor. The authors simulated the preparation and measurement of anyons, leveraging dynamic string-net preparation strategies to realize the non-trivial topological quantum states. Specifically, the paper focuses on the relationship between string-net wavefunctions and chromatic polynomials and employs quantum computing to estimate these polynomials. The authors utilized advanced quantum circuits and error suppression techniques to measure and manipulate Fib-SNC. They provide a detailed theoretical framework validated through extensive experimentation, showcasing the practicality and effectiveness of their methods. The foremost contribution of this work is the innovative application of quantum computing to estimate chromatic polynomials. This novel approach represents a significant advancement in addressing the potentially #P-hard problems using quantum computing techniques, thereby demonstrating the potential for quantum advantage in this domain. Moreover, the manuscript provides a comprehensive account of sophisticated error suppression and quantum gate optimization strategies. The authors employ auxiliary qubits to reduce the depth of Toffoli gates from 63 to 18, substantially improving the fidelity and reliability of the experimental results. We judge this work an interesting and timely significant contribution along the direction of quantum simulation. We believe the demonstration of braiding Fibonacci anyons is an important breakthrough and this work is well-qualified for publication in Nature Communications.

Our major concern is about the presentation, as detailed in the following. We are happy to recommend its acceptance in Nature Communications, after authors carefully addressed the following comments.

Response: We thank the referee for the enthusiastic endorsement of our work, careful manuscript reading, and detailed suggestions.

(1) **Writing Quality:** The main text focuses excessively on experimental details. It dedicates an extensive amount of text to the experimental details surrounding the preparation and manipulation of states, which overshadows the discussion of the novel contributions. The frequent use of qubit indices (e.g., Q1, Q2, etc.) further reduces the readability and detracts from the primary innovations of the work.

Response: We thank the referee for calling us to improve upon the discussion of the novel contributions. Per referee's suggestions, we moved the description of the experimental protocol for Figure 2 that was heavy on the use of the

qubit indices to a new Methods section. In addition, we revised paragraphs [9], [10], and [12] to improve the flow of the narrative and focus on the novel aspects of the creation of the Fibonacci anyons and certification of their types.

(2) Additionally, the manuscript suffers from poor formatting, including inconsistencies in footnote symbols. A critical section of the supplementary material is missing (page 13, containing equations C10 to C13 and figure 4), which makes it hard to understand the key process of estimating the chromatic polynomials using quantum computing. This omission is particularly detrimental as it obscures the main advantage of the paper.

Response: We thank the referee for catching these important issues. We have carefully reviewed the manuscript and corrected all the inconsistencies in the footnote symbols that we could identify.

We also sincerely apologize for the missing page in the supplementary material. It appears there was a technical issue during the submission process that led to the omission of a critical page containing Equations C10–C13 and Figure 4. This was certainly unintentional, and we fully agree that this page is essential to understanding the quantum estimation process and the key advantage of the work.

We have double-checked and ensured that the new version of the supplementary material now includes the complete content, including the previously missing page with all equations and figures intact. We hope this resolves the confusion and greatly improves clarity for the reader.

(3) Furthermore, figures in the main text, such as figure 2b, have confusing labeling (e.g., labels 2 and 6 for the qubits), and the lines and qubits in Figure 3 appear disjointed.

Response: We thank the referee for this helpful observation regarding the clarity of labeling and graphical continuity in the figures. In response, we have carefully reviewed and updated Figure 2 to address the potential confusion in labeling. Now, we have moved the qubit labels on top of the corresponding qubits to ensure the clarity of the spatial and logical relationships amongst the qubits, the graph, and the string paths. We have also revised the figure caption to explain the roles of each labeled qubit and their positions in the graph more explicitly.

For Figure 3, we have revised how we describe the five-qubit F -moves. Instead of using isolated purple edges, which had potentially contributed to the confusion, we now use the orange patches to indicate the groups of five qubits that undergo the F -moves, ensuring the visual clarity of the circuit logic. We've also made sure the qubits and the underlying graphs are properly connected. Additionally, we've updated the figure caption to clarify the graphical presentation of the anyons and the qubits. We hope these updates significantly improve readability and interpretation for the reader.

(4) Lack of Emphasis on Key Contribution: The primary distinction between this work and the previous study (Ref. [26]) is the estimation of the chromatic polynomials using quantum computing. However, the manuscript does not adequately emphasize this novel contribution. The significance and innovation of this approach are not sufficiently highlighted, making it difficult to appreciate the advancement it offers over existing methods. A more focused discussion on how this work advances the field relative to past research would strengthen the manuscript and make its contributions clearer to the readers.

Response: We thank the referee for urging us to emphasize the significance and innovation of the chromatic polynomial estimation. We revised the text throughout to follow the referee's advice. In particular, we have substantially revised paragraphs [15-19] and added specific polynomials evaluated in Fig 4 through equations (4-9) for concreteness. We hope these updates put adequate emphasis on the significance of this most notable result of our work.

(5) We are concerned about the efficiency of this method for larger systems. Even though the authors employ the approach of sampling the isomorphism class $[G]$ instead of the graph G . Additionally, in the second paragraph of page 4, the authors state, "Strikingly, the experimental distribution shows dramatic suppression of the bitstrings violating the branching rules." However, it is difficult to understand how this conclusion is drawn from the preceding sentence, "Of these, only 47 bitstrings respect the branching rules of the Fib-SNC." The use of the word "only" suggests a small number, which seems to contradict the notion of dramatic suppression of invalid bitstrings mentioned earlier.

Response: We tried to clarify the text by revising the end of the paragraph [17]. 47 bitstrings respect the branching rule, and Fig 4g shows the rule-violating bitstrings (outside first 47) are generally immensely suppressed in their amplitudes. There are a total of 2^{11} bitstrings since the data includes ancilla qubits.

Report of Referee 2

However, the title of the paper is not accurate to me. The paper deals with a single classically-hard problem, the chromatic polynomial evaluation. The paper is not really about topological quantum computing other than simulating

anyon states which are different from the anyons in topological phases which are protected by a gap. I suggest using the title from the arXiv version ("Realizing string-net condensation: Fibonacci anyon braiding for universal gates and sampling chromatic polynomials") or something similar. I recommend it for publication in the journal as long as the title issue is resolved.

Response: We thank the referee for enthusiastically recommending the manuscript's publication. Per the referee's recommendation, we have changed the title.

Report of Referee 3

This is excellent work and deserves publication in a high-profile journal. The research is technically advanced, and the authors have made a commendable effort to make it accessible to a broader audience. I have little to add and recommend the publication as is.

I believe mentioning Ref. 26 in the acknowledgments is sufficient. Since the work appears to have been completed around the same time, multiple references in the main text could potentially confuse the reader.

While I appreciated the opening of the abstract, I should point out that even the vacuum of a harmonic oscillator is not truly empty, as it exhibits quantum fluctuations. The key distinction, in my view, lies in the complexity and many-body nature of topologically ordered vacua.

Response: Thank you for your enthusiastic endorsement of the work. We removed the first sentence of the abstract and improved what used to be the second sentence to emphasize the complexity and many-body nature of the topologically ordered vacua.

Report of Referee 4

The authors use IBM quantum processors to analyse topological behaviour in and around the Fibonacci string-net ground state. In particular, they confirm the intriguing property that the ground-state wavefunction amplitudes depend on the chromatic polynomial. They also show to good accuracy that the anyonic excitations exhibit the non-Abelian braiding properties expected.

Whereas I tend to be skeptical of papers utilising quantum computers to confirm previously understood physics, I think here the physics is rather novel and far from obvious. Moreover, the work done in this paper really pushes the limit of what is experimentally possible – I would have guessed we would need to wait some years before such intricate properties of Fibonacci anyons could be analysed in a quantum computer. I thus recommend publication of this paper in Nature Communications.

Response: We thank the referee for carefully reading the manuscript, making detailed suggestions, and enthusiastically recommending that our work be published in a high-profile journal.

The authors do need to make some small revisions.

– The first sentence of the abstract is a bit gratuitous, since I don't believe any good physicists are unaware that the ground state of a quantum system can have non-trivial properties. (Even though particle theorists use the word "vacuum" to mean "ground state", they know it's not trivial.) So I would delete that sentence and not say "topologically ordered vacuum" in the next, but rather "topologically ordered ground state"

Response: We removed the first sentence of the abstract and improved what used to be the second sentence to emphasize the complexity and many-body nature of the topologically ordered vacua.

– The authors need to make clear that they are evaluating the chromatic polynomial at a particular number of colours, not in general. They say this in some places, but not everywhere (e.g. in the overview before equation (1) and in their summary).

Response: Thanks for pointing that out. We have now added the description in the following sentence in paragraph [5] "Furthermore, DSNP allows us to make the first steps towards scaling up the Fib-SNC to estimate the chromatic polynomial at golden ratio $\phi + 2$." and the following sentence in paragraph [19] "Hence it is possible that the estimation of chromatic polynomial at $\phi + 2$ from quantum sampling can be more efficient than from classical algorithms for intermediate system size (for $O(100)$ to $O(1000)$ qubits)."

– It's not clear what a "tail qubit" is, unless one reads the supplemental information. The authors should add a sentence explaining in the main paper.

Response: We have added an explanation in the following sentence in paragraph [7] “Furthermore, to create anyons while allowing for detection and correction of local errors, we follow the ‘tail anyon’ strategy [20] that traps an end of an open string to the ‘tail qubit’ located on a dangling edge inside a plaquette.”

– The pdf of the supplemental information is missing page 6 (at least the one I received).

Response: We thank the referee for pointing that out. We have fixed this issue in the updated PDF file for the supplemental materials. Now it does not have missing page.

In this work, the authors explore the quantum simulation of Fibonacci string-net condensation (Fib-SNC) and demonstrate its potential applications in topological quantum computing. The research is conducted through a series of experiments on the IBM Heron quantum processor. The authors simulated the preparation and measurement of anyons, leveraging dynamic string-net preparation strategies to realize the non-trivial topological quantum states. Specifically, the paper focuses on the relationship between string-net wavefunctions and chromatic polynomials and employs quantum computing to estimate these polynomials. The authors utilized advanced quantum circuits and error suppression techniques to measure and manipulate Fib-SNC. They provide a detailed theoretical framework validated through extensive experimentation, showcasing the practicality and effectiveness of their methods.

The foremost contribution of this work is the innovative application of quantum computing to estimate chromatic polynomials. This novel approach represents a significant advancement in addressing the potentially #P-hard problems using quantum computing techniques, thereby demonstrating the potential for quantum advantage in this domain. Moreover, the manuscript provides a comprehensive account of sophisticated error suppression and quantum gate optimization strategies. The authors employ auxiliary qubits to reduce the depth of Toffoli gates from 63 to 18, substantially improving the fidelity and reliability of the experimental results.

We judge this work an interesting and timely significant contribution along the direction of quantum simulation. We believe the demonstration of braiding Fibonacci anyons is an important breakthrough and this work is well-qualified for publication in Nature Communications. Our major concern is about the presentation, as detailed in the following. We are happy to recommend its acceptance in Nature Communications, after authors carefully addressed the following comments.

Main Concerns:

Writing Quality: The main text focuses excessively on experimental details. It dedicates an extensive amount of text to the experimental details surrounding the preparation and manipulation of states, which overshadows the discussion of the novel contributions. The frequent use of qubit indices (e.g., Q1, Q2, etc.) further reduces the readability and detracts from the primary innovations of the work. Additionally, the manuscript suffers from poor formatting, including inconsistencies in footnote symbols. A critical section of the supplementary

material is missing (page 13, containing equations C10 to C13 and figure 4), which makes it hard to understand the key process of estimating the chromatic polynomials using quantum computing. This omission is particularly detrimental as it obscures the main advantage of the paper. Furthermore, figures in the main text, such as figure 2b, have confusing labeling (e.g., labels 2 and 6 for the qubits), and the lines and qubits in figure 3 appear disjointed.

Lack of Emphasis on Key Contribution: The primary distinction between this work and the previous study (Ref. [26]) is the estimation of the chromatic polynomials using quantum computing. However, the manuscript does not adequately emphasize this novel contribution. The significance and innovation of this approach are not sufficiently highlighted, making it difficult to appreciate the advancement it offers over existing methods. A more focused discussion on how this work advances the field relative to past research would strengthen the manuscript and make its contributions clearer to the readers.

We also have some questions on the technical level:

The authors claimed that measuring the bitstring counts corresponding to graphs topologically equivalent to G can address the problem, which is expected to be #P-hard, of estimating the chromatic polynomial $\chi(\hat{G}, \phi + 2)$. We are concerned about the efficiency of this method for larger systems. Even though the authors employ the approach of sampling the isomorphism class $[G]$ instead of the graph G .

Additionally, in the second paragraph of page 4, the authors state, “Strikingly, the experimental distribution shows dramatic suppression of the bitstrings violating the branching rules.” However, it is difficult to understand how this conclusion is drawn from the preceding sentence, “Of these, only 47 bitstrings respect the branching rules of the Fib-SNC.” The use of the word “only” suggests a small number, which seems to contradict the notion of dramatic suppression of invalid bitstrings mentioned earlier.

Report for

Towards classically-hard problems and universal topological quantum computation
through realizing Fibonacci string-net condensate

This paper explores the experimental implementation of the Fibonacci string-net model on two advanced quantum processors, the 27-qubit IBM Falcon and the 133-qubit IBM Heron. String-net models are a class of exactly solvable lattice models that give rise to two-dimensional topological phases of matter, characterized by quasi-particle excitations known as anyons. Topological phases featuring non-Abelian anyons are of particular importance for fault-tolerant topological quantum computing. When the Fibonacci anyon theory serves as the input to the string-net model, the resulting excitations consist of two time-reversed copies of the Fibonacci theory. The Fibonacci anyon theory, along with its double, holds special significance as it represents the simplest known anyon theories that are universal for quantum computation via braiding alone. Although the Fibonacci anyon is hypothesized to exist in certain fractional quantum Hall states, experimental confirmation remains elusive.

With the advent of increasingly powerful quantum computers, it has become feasible to simulate topological states directly on these platforms. Notable achievements in this area include the realization of the toric code by Google's team, the simulation of non-Abelian DD_4 anyon states by Quantinuum, and, more recently, the work published in Nature Physics (vol. 20, pp. 1469–1475, 2024) by a Chinese research team on simulating Fibonacci string-net states. For brevity, this last work will be referred to as the Z paper.

The current paper also explores the simulation of the Fibonacci string-net on quantum processors, demonstrating ground state preparation, anyon creation, and anyon braiding on superconducting quantum platforms, similar to the Z paper. The current paper appeared in the arXiv about 2.5 months later than the Z paper, hence it is reasonable to speculate that these two are independent of each other. Moreover, this work makes several novel contributions that distinguish it from the Z paper.

First, unlike the Hamiltonian-based method used in the Z paper, the current study introduces a dynamic string-net preparation approach on graphs, utilizing minimalistic string-net structures as building blocks. This approach offers the flexibility of not requiring a fixed lattice, enabling the dynamic construction of ground states. Additionally, it eliminates the need for 12-spin measurements of plaquette operators, typically required in the conventional honeycomb lattice setup.

Second, the paper addresses a classically challenging problem: the evaluation of the chromatic polynomial on graphs. The string-net ground state amplitudes are known to correspond to the chromatic polynomial evaluated at the specific value $\phi + 2$, where ϕ is the golden ratio. By realizing the Fibonacci ground state on quantum processors and sampling the resulting amplitudes, the study provides estimates of the chromatic polynomial for various graphs.

Overall, the paper demonstrates a proof-of-principle for simulating the Fibonacci string-net and leveraging it to address a classically hard problem. Compared to the Z paper, the approach presented here appears more scalable. The technical content of the paper appears sound. However, the title of the paper is not accurate to me. The paper deals with a single classically-hard problem, the chromatic polynomial evaluation. The paper is not really about topological quantum computing other than simulating anyon states which are different from

the anyons in topological phases which are protected by a gap. I suggest using the title from the arXiv version ("Realizing string-net condensation: Fibonacci anyon braiding for universal gates and sampling chromatic polynomials") or something similar. I recommend it for publication in the journal as long as the title issue is resolved.